# VAE APPROXIMATION ERROR: ELBO AND EXPONENTIAL FAMILIES

**Alexander Shekhovtsov**
Czech Technical University in Prague
shekhole@fel.cvut.cz

**Dmitrij Schlesinger**
Dresden University of Technology
Dmytro.Shlezinger@tu-dresden.de

**Boris Flach**
Czech Technical University in Prague
flachbor@fel.cvut.cz

## ABSTRACT

The importance of Variational Autoencoders reaches far beyond standalone generative models — the approach is also used for learning latent representations and can be generalized to semi-supervised learning. This requires a thorough analysis of their commonly known shortcomings: posterior collapse and approximation errors. This paper analyzes VAE approximation errors caused by the combination of the ELBO objective and encoder models from conditional exponential families, including, but not limited to, commonly used conditionally independent discrete and continuous models. We characterize subclasses of generative models consistent with these encoder families. We show that the ELBO optimizer is pulled away from the likelihood optimizer towards the consistent subset and study this effect experimentally. Importantly, this subset can not be enlarged, and the respective error cannot be decreased, by considering deeper encoder/decoder networks.

## 1 INTRODUCTION

Variational autoencoders (VAE, Kingma & Welling, 2014; Rezende et al., 2014) strive at learning complex data distributions $p_d(x)$, $x \in \mathcal{X}$ in a generative way. They introduce latent variables $z \in \mathcal{Z}$ and model the joint distribution as $p_\theta(x \,|\, z)p(z)$, where $p(z)$ is a simple distribution which is usually assumed to be known. The conditional distribution $p_\theta(x \,|\, z)$, called *decoder*, is modeled in terms of a deep network parametrized by $\theta \in \Theta$. Models defined in this way allow to sample from $p_\theta(x) = \mathbb{E}_{p(z)} p_\theta(x \,|\, z)$ easily, however at the price that computing the *posterior* $p_\theta(z \,|\, x) = p_\theta(x \,|\, z)p(z)/p_\theta(x)$ is usually intractable. To handle this problem, VAE approximates the posterior $p_\theta(z \,|\, x)$ by an amortized inference *encoder* $q_\phi(z \,|\, x)$ parametrized by $\phi \in \Phi$. Given the empirical data distribution $p_d(x)$, the model is learned by maximizing the evidence lower bound (ELBO) of the data log-likelihood $L(\theta) = \mathbb{E}_{p_d} \log p_\theta(x)$. It can be expressed in the following two equivalent forms:

$$L_B(\theta, \phi) = \mathbb{E}_{p_d}\big[\mathbb{E}_{q_\phi} \log p_\theta(x \,|\, z) - D_{\mathrm{KL}}(q_\phi(z \,|\, x) \,\|\, p(z))\big] \tag{1a}$$

$$= L(\theta) - \mathbb{E}_{p_d}\big[D_{\mathrm{KL}}(q_\phi(z \,|\, x) \,\|\, p_\theta(z \,|\, x))\big]. \tag{1b}$$

The first form allows for stochastic optimization of ELBO while the second form shows that the gap between log-likelihood and ELBO is exactly the mismatch between the encoder and the posterior.

VAEs constitute a powerful deep learning extension of the expectation-maximization (EM) approach to handle latent variables. They are useful not only as generative models but also, *e.g.*, in semi-supervised learning (Kingma et al., 2014; Mattei & Frellsen, 2019). Furthermore the encoder part constructs an efficient embedding of the data in the latent space, useful in many applications. The outreach of the VAE approach requires therefore a careful empirical and theoretical analysis of the problems and trade offs involved. The most important ones are (i) posterior collapse (He et al., 2019; Lucas et al., 2019; Dai et al., 2018; Dai & Wipf, 2019; Dai et al., 2020) and (ii) approximation errors caused by an inappropriate choice of the encoder family.

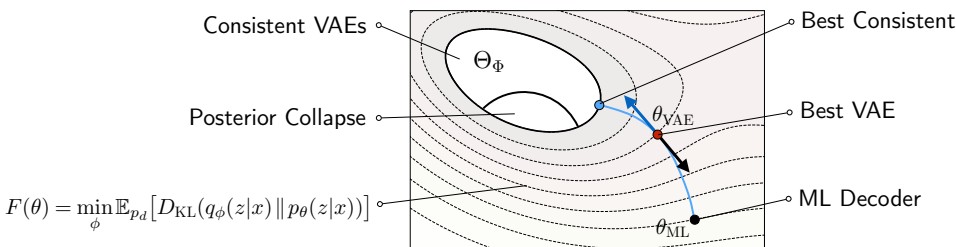

Figure 1: Diagram of the VAE trade-off. The optimal solution $\theta_{\mathrm{VAE}}$ is "in between" the maximum likelihood solution $\theta_{\mathrm{ML}}$ and the best solution in the class $\Theta_\Phi$ of consistent VAEs, where the posterior approximation error function $F(\theta)$ vanishes. We give an explicit characterization of this consistent set. At $\theta_{\mathrm{VAE}}$ there is a balance between the gradient of $-F$ (blue arrow) and the gradient of the data log-likelihood (black arrow).

The VAE approximation error has been studied (*e.g.*, Cremer et al. 2018; Hjelm et al. 2016; Kim et al. 2018) so far mainly empirically. The problem also occurs and is well-recognized in the context of variational inference and variational Bayesian inference, where the target posterior distribution is expected to be complex. It is commonly understood, that the mean field approximation of $p_\theta(z\,|\,x)$ by $q_\phi(z\,|\,x)$ in (1b) significantly limits variational Bayesian inference. In contrast, in VAEs, the decoder may adopt to compensate for the chosen encoder family. The effect of this coupling, we believe, is not fully understood. The phenomenon of decoder adopting to the posterior was experimentally observed, *e.g.*, by Cremer et al. (2018, Section 5.4), noting that the approximation error is often dominated by the amortization error. Turner & Sahani (2011, Sec. 1.4) analytically show for linear state space models that simpler variational approximations (such a mean-field) can lead to less bias in parameter estimation than more complicated structured approximations. Similarly, Shu et al. (2018) view the VAE objective as providing a regularization and show that making the amortized inference model smoother, while increasing the amortization gap, leads to a better generalization.

The common (empirical) understanding of the importance of the gap between the approximate and the true posterior has led to many generalizations of standard VAEs, which achieve impressive practical results, notably, tighter bounds using importance weighting (Burda et al., 2016; Nowozin, 2018), encoders employing normalizing flows (Rezende & Mohamed, 2015; Kingma et al., 2016), hierarchical and autoregressive encoders (Vahdat & Kautz, 2020; Sønderby et al., 2016; Ranganath et al., 2016), MRF encoders (Vahdat et al., 2020) and more. While these extensions mitigate the posterior mismatch problem, they often come at a price of a more difficult training and more expensive inference. Furthermore, simpler encoders may be of practical interest. Burda et al. (2016, Appendix C) illustrates that IWAE approximate posteriors are less regular and more spread out. In contrast, factorized encoders provide simple embeddings useful for downstream tasks such as semantic hashing (Chaidaroon & Fang, 2017).

The aim of this paper is to study the approximation error of VAEs and its impact on the learned decoder. We consider a setting that generalizes many common VAEs, in particular popular models where encoder and decoder are conditionally independent Bernoulli or Gaussian distributions: we assume that both decoder and encoder are conditional exponential families. We identify the subclass of generative models where the encoder can model the posterior exactly, referred to as *consistent* VAEs. We give a characterization of consistent VAEs revealing that this set in fact does not depend on the complexity of the involved neural networks. We further show that the ELBO optimizer is pulled towards this set away from the likelihood optimizer. Specializing the characterization to several common VAE models, we show that the respective consistent models turn out to be RBM-like in many cases. We experimentally investigate the detrimental effect in one case and show that a simpler but more consistent VAE can perform better in the other.

## 2 PROBLEM STATEMENT

We adopt the following notion of approximation error. Consider a generative model class $\mathcal{P}_\Theta = \{p_\theta(x, z) \mid \theta \in \Theta\}$, the encoder class $\mathcal{Q}_\Phi = \{q_\phi(z\,|\,x) \mid \phi \in \Phi\}$ and the data distribution $p_d(x)$. The maximum likelihood generative model is given by $\theta_{\mathrm{ML}} \in \mathrm{argmax}_{\theta \in \Theta}\, \mathbb{E}_{p_d(x)} \log p_\theta(x)$. For a decoder with parameters $\theta$ we define its *approximation error* as the likelihood difference $L(\theta_{\mathrm{ML}}) -$

$L(\theta)$. Respectively, the *VAE approximation error* is defined for a given $\theta$ as:

$$L(\theta_{\mathrm{ML}}) - \max_\phi L_B(\theta, \phi) \geq L(\theta_{\mathrm{ML}}) - L(\theta). \tag{2}$$

In order for this error to become zero, two conditions are necessary and sufficient:

- Parameters $(\theta, \phi)$ must be optimal for the ELBO objective.
- ELBO must be *tight* at $(\theta, \phi)$, *i.e.*, $L_B(\theta, \phi) = L(\theta)$.

Assuming that the optimality can be achieved, we study the non-tightness gap $L(\theta) - L_B(\theta, \phi)$. From (1b) it expresses as $\mathbb{E}_{p_d}\big[D_{\mathrm{KL}}(q_\phi(z\,|\,x) \,\|\, p_\theta(z\,|\,x))\big]$. It follows that ELBO is tight at $(\theta, \phi)$ iff $q_\phi(z\,|\,x) \equiv p_\theta(z\,|\,x)$. Hence, we define the *consistent set* $\Theta_\Phi \subseteq \Theta$ as the subset of distributions $p_\theta(x, z)$ whose posteriors are in $\mathcal{Q}_\Phi$, *i.e.*,

$$\Theta_\Phi = \big\{\theta \in \Theta \;\big|\; \exists \phi \in \Phi : q_\phi(z\,|\,x) \equiv p_\theta(z\,|\,x)\big\}. \tag{3}$$

The KL-divergence in the ELBO objective (1b) can vanish only if $\theta \in \Theta_\Phi$. If the likelihood maximizer $\theta_{\mathrm{ML}}$ is not contained in $\Theta_\Phi$, then this KL-divergence pulls the optimizer towards $\Theta_\Phi$ and away from $\theta_{\mathrm{ML}}$ as illustrated in Fig. 1.

We characterize the consistent set $\Theta_\Phi$, on which the bound is tight, and show that this set is quite narrow and does not depend on the complexity of the encoder and decoder networks beyond simple 1-layer linear mappings of sufficient statistics.

## 3 THEORETICAL ANALYSIS

We consider a general class of VAEs, where both encoder and decoder are defined as exponential families. This class includes many common models, in particular Gaussian VAEs and Bernoulli VAEs with conditional independence assumptions, but also more complex ones, *e.g.*, where the encoder is a conditional random field (Vahdat et al., 2020)[1].

**Assumption 1** (Exponential family VAE). *Let $\mathcal{X}$ and $\mathcal{Z}$ be sets of observations and latent variables, respectively. We consider VAE models defined by*

$$p_\theta(x\,|\,z) = h(x)\exp\big[\langle\nu(x), f_\theta(z)\rangle - A(f_\theta(z))\big] \tag{4a}$$

$$q_\phi(z\,|\,x) = h'(z)\exp\big[\langle\psi(z), g_\phi(x)\rangle - B(g_\phi(x))\big], \tag{4b}$$

*where $\nu\colon \mathcal{X} \to \mathbb{R}^n$ and $\psi\colon \mathcal{Z} \to \mathbb{R}^m$ are fixed sufficient statistics of dimensionality $n$ and $m$; $f_\theta\colon \mathcal{Z} \to \mathbb{R}^n$ and $g_\phi\colon \mathcal{X} \to \mathbb{R}^m$ are the decoder, resp., encoder, networks with learnable parameters $\theta$, resp. $\phi$; $h\colon \mathcal{X} \to \mathbb{R}_+$, $h'\colon \mathcal{Z} \to \mathbb{R}_+$ are strictly positive base measures and $A$, $B$ denote the respective log-partition functions.*

Notice that this assumption imposes no restrictions on the nature of random variables $x$ and $z$. They can be discrete or continuous, univariate or multivariate. Similarly, it imposes no restrictions on the complexity of the decoder and encoder networks $f_\theta(z)$ and $g_\phi(x)$.

**Characterization of the consistent set.** In the first step of our analysis, we investigate the conditions under which the approximation error of an exponential family VAE can be made exactly zero. As discussed above, a tight VAE $(\theta, \phi)$ must satisfy $\forall(x,z)\ \ q_\phi(z\,|\,x) = p_\theta(z\,|\,x)$, which leads to the following theorem.

**Theorem 1.** *The consistent set $\Theta_\Phi$ of an exponential family VAE is given by decoders of the form*

$$p(x\,|\,z) = h(x)\exp\big[\langle\nu(x), W\psi(z)\rangle + \langle\nu(x), u\rangle - A(z)\big], \tag{5}$$

*where $W$ is a $n \times m$ matrix and $u \in \mathbb{R}^n$. Moreover, the corresponding encoders have the form*

$$q(z\,|\,x) = h'(z)\exp\big[\langle\psi(z), W^T\nu(x)\rangle + \langle\psi(z), v\rangle - B(x)\big], \tag{6}$$

*where $v \in \mathbb{R}^m$.*

---

[1]Notice, however, that this class does not include VAEs with advanced encoder families like normalizing flows, hierarchical and autoregressive encoders.

This is a direct consequence of a theorem by Arnold & Strauss (1991) (see Appendix A.1 for more details). For a tight VAE, Theorem 1 states that the decoder and encoder are *generalized linear models* (GLMs) (5) and (6) with the interaction between $x$ and $z$ parametrized by a matrix $W$ and two vectors $u, v$ instead of the (complex) neural networks with parameters $\theta, \phi$. The corresponding joint probability distribution takes the form of an EF Harmonium (Welling et al., 2005):

$$p(x, z) = h(x)h'(z) \exp\big(\langle \nu(x), W\psi(z)\rangle + \langle \nu(x), u\rangle + \langle \psi(z), v\rangle - A\big). \tag{7}$$

**Corollary 1.** *The subset $\Theta_\Phi$ of consistent models can not be enlarged by considering more complex encoder networks $g(x)$, provided that the affine family $W^\mathsf{T}\nu(x)$ can already be represented.*

**Corollary 2.** *Let the decoder network family be affine in $\psi(z)$, i.e., $f(z) = W\psi(z) + a$ and let the encoder network family $g(x)$ include at least all affine maps $V\nu(x) + b$. Then any global optimum of ELBO attains a zero approximation error.*

**VAE can escape consistency when it degenerates to a flow.** In practice, VAE models with rich decoders are almost never tight. It is therefore natural to ask, whether a small VAE posterior mismatch error implies closeness of the optimal decoder to some decoder in the consistent set.

**Definition 1.** *A VAE $(p_\theta, q_\phi)$ is $\varepsilon$-tight for some $\varepsilon > 0$ if $\mathbb{E}_{p_d(x)}[D_{\mathrm{KL}}(q_\phi(z|x)\|p_\theta(z|x))] \leq \varepsilon$.*

It turns out that this definition allows a VAE to approach tightness while not approaching consistency. In the continuous case an example satisfying $\varepsilon$-tightness with non-linear decoder follows from Dai & Wipf (2019, Theorem 2). They show, for a class of Gaussian VAEs with general neural networks $f_\theta, g_\phi$, that it is possible to build a sequence of network parameters $\theta_t, \phi_t$ with the following properties: i) the target distribution is approximated arbitrary well, ii) the posterior mismatch $D_{\mathrm{KL}}(q_{\phi_t}(z|x)\|p_{\theta_t}(z|x))$ approaches zero and iii) both the encoder and decoder approach deterministic mappings. The VAE thus approaches a flow model (or invertible neural network) between the data manifold and a subspace of the latent space (Dai & Wipf, 2019). Clearly, in a general case the flow must be non-linear. A similar case can be made for discrete variables, see Example A.1.

**Non-deterministic nearly-tight VAEs approach consistency.** We would however argue that the mode where the decoder and encoder are nearly-deterministic is not a natural VAE solution. By making additional assumptions, excluding such deterministic solutions, and restricting ourselves to the finite space in order to simplify the analysis, we can show that an $\varepsilon$-tight VAE does indeed approach an EF-Harmonium.

**Theorem 2.** *Let $(p_\theta, q_\phi)$ be an exponential family VAE (Assumption 1) on a discrete space $\mathcal{X} \times \mathcal{Z}$ with encoder $q_\phi(z|x)$ and decoder posterior $p_\theta(z|x)$ both bounded from below by $\alpha > 0$. If the VAE is $\varepsilon$-tight, then there exists a matrix $W \in \mathbb{R}^{n,m}$ and vectors $u \in \mathbb{R}^n$, $v \in \mathbb{R}^m$ such that the joint model implied by the decoder $p_\theta(x, z) = p_\theta(x|z)p(z)$ can be approximated by an unnormalized EF Harmonium*

$$\tilde{p}(x, z) = h(x)h'(z) \exp(\langle \nu(x), W\psi(z)\rangle + \langle \nu(x), u\rangle + \langle v, \psi(z)\rangle + c) \tag{8}$$

*with the error bound*

$$\mathbb{E}_{p_d(x)}\Big[\big(\log p_\theta(x, z) - \log \tilde{p}(x, z)\big)^2\Big] \leq \tfrac{\varepsilon}{2\alpha^2} + o(\varepsilon) \quad \forall z \in \mathcal{Z}. \tag{9}$$

The proof is given in Appendix A.3. In this theorem the function $\tilde{p}(x, z)$ is non-negative but does not necessarily satisfy the normalization constraint of a density. Re-normalizing it by adjusting $c$ in (8) may break the approximation guarantee. Nevertheless, if $\varepsilon$ is small enough and, *e.g.*, the data distribution is non-negative on the whole $\mathcal{X}$, we expect it to approach a density, in particular to recover the result in Theorem 1 in the limit. Note that the theorem does not make any assumptions about optimality of $(\theta, \phi)$, *i.e.*, it describes all models in the vicinity of the consistent set in Fig. 1.

## 3.1 CASES ANALYSIS

This subsection gives a detailed analysis of consistent VAE models in several concrete cases of practical interest.

**Diagonal Gaussian VAE**   Let us consider a Gaussian VAE, as commonly applied to image generation (*e.g.*, Dai & Wipf (2019)). Let $\mathcal{X} = \mathbb{R}^n$, $\mathcal{Z} = \mathbb{R}^m$, $p(x|z) = \mathcal{N}(x|\mu_d(z), \sigma_d^2 I)$,

$q(z \,|\, x) = \mathcal{N}(z \,|\, \mu_e(x), \mathrm{diag}(\sigma_e^2(x)))$, where $\mu_d$, $\mu_e$ and $\sigma_e$ are neural networks and $\sigma_d$ is a common pixel observation noise parameter. The decoder has minimal sufficient statistics $\nu(x) = x$ and base measure $h(x) = \mathcal{N}(x \,|\, 0, \sigma_d^2 I)$. The encoder has minimal sufficient statistics $\psi(z) = (z, z^2)$, where the square is coordinate-wise. Theorem 1 implies that a tight optimal VAE has the joint model

$$p(x, z) \propto h(x) \exp\big[\langle x, Wz + Vz^2 + a \rangle + \langle b, z \rangle + \langle c, z^2 \rangle\big] \qquad (10)$$

for some matrices $W$, $V$ and vectors $a, b, c$. Furthermore, the integral over $z$ must be finite for all $x$ and therefore $x^\mathsf{T} V + c < 0$ must hold for all $x \in \mathbb{R}^n$. This is possible only if $V = 0$ and $c < 0$. The joint distribution is therefore a multivariate Gaussian and the same holds for its marginal $p(x)$. The neural network $\mu_d(z)$ must degenerate to $\mu_d(z) = \sigma_d^2 \cdot (Wz + a)$ and the two neural networks for the encoder to $\sigma_e^2(x) = -1/2c$ and $\mu_e(x) = -(W^\mathsf{T} x + b)/2c$, where divisions are coordinate-wise. VAEs with such simplified, linear Gaussian encoder-decoder pairs, called "linear VAEs" (Lucas et al., 2019) are known to be consistent and to match the probabilistic PCA model (Dai et al., 2018; Lucas et al., 2019). In this context, our Corollary 2 is a generalization of (Lucas et al., 2019, Lemma 1) showing consistency of linear VAEs, to decoders in any exponential family with natural parameters being a linear mapping of any fixed lifted latent representation $\psi(z)$.

We argue that a joint Gaussian model is too simplistic to generate complex data such as realistic images and that in this case the VAE error is detrimental. In Section 4.2 we experimentally confirm that optimizing ELBO for a general decoder network $\mu_d$ causes qualitative and quantitative degradation relative to the ML decoder. Note that if we allowed $\sigma_d$ to be dependent on $z$, the resulting joint statistics in $\nu \otimes \psi$ would include terms $x^2 z$, $x^2 z^2$. The joint distribution would not be Gaussian and may be in fact multi-modal (see Anil Bhattacharayya's distribution in Arnold et al. 2001).

**Bernoulli-MRF VAE** Vahdat et al. (2020) proposed to consider encoders in the Markov Random Field (MRF) family, in particular encoders of the form $q(z \,|\, x) \propto \exp(\langle z^1, V(x) z^2 \rangle + \langle b^1(x), z^1 \rangle + \langle b^2(x), z_2 \rangle)$, where $z^1$, $z^2$ are two groups of latent variables and interaction weights $V$, $b^1$, $b^2$ are computed by the encoder network. In this case $q(z \,|\, x)$ is itself a (conditional) RBM. While evaluating $q(z \,|\, x)$ is difficult, MCMC sampling is efficient. We assume binary observations $x$ and a conditionally independent Bernoulli decoder family as above. The decoder thus has sufficient statistics $\nu = x$ and the encoder has $\psi = (z^1, z^2, z^1 \otimes z^2)$. Introducing homogeneous constant components $x_0 = z_0^1 = z_0^2 = 1$, the family of consistent joint distributions can be compactly described as

$$p(x, z) = \exp\big[\textstyle\sum_{i,j,k} W_{i,j,k} \, x_i \, z_j^1 \, z_k^2\big], \qquad (11)$$

where the summation in all indices starts from 0 and $-W_{0,0,0}$ is the log-partition function. This joint model is a higher order MRF with the highest order potentials given by cubic monomials.

Standard Bernoulli VAEs are a special case of the Bernoulli-MRF model, obtained when the interaction weights $V$ are zero. The joint distribution of such tight optimal VAEs takes the form

$$p(x, z) = \tfrac{1}{c} \exp(x^\mathsf{T} W z + u^\mathsf{T} x + v^\mathsf{T} z), \qquad (12)$$

which is a restricted Boltzmann machine (RBM). Since RBMs are well known for being useful in many applications (dimensionality reduction, collaborative filtering, feature learning, topic modeling), we hypothesize that they can make a good baseline for Bernoulli VAEs and furthermore that the effect of pulling the VAE solution towards an RBM may be benign in case of insufficient data. For example IWAE test likelihood in (Burda et al., 2016) is worse than that of an RBM (Burda et al., 2015) on the Omniglot dataset. Furthermore, debiasing of IWAE (Nowozin, 2018) does not improve test likelihood in many cases.

**Bernoulli VAE for Semantic Hashing** One important application of Bernoulli VAEs is the *semantic hashing* problem, initially proposed and modeled with RBMs (Salakhutdinov & Hinton, 2009). The problem is to assign to each document / image a compact binary latent code that can be used for quick retrieval by the nearest neighbor search. We will detail now a more recent VAE model for text documents (Chaidaroon & Fang, 2017; Shen et al., 2018) and show that it can be tight only in a full posterior collapse. We correct the encoder so as to allow a larger consistent set and observe that the resulting consistent joint distribution forms a multinomial-Bernoulli RBM.

Let $x \in \mathbb{N}^K$ be word counts in a document with words from a dictionary of size $K$. Let $z \in \{0, 1\}^m$ be a binary latent code. Let $l = \sum_k x_k$ denote the document's length. We assume that the document

length is independent of the latent topic and its distribution $p(l)$ can be learned separately (*e.g.*, a log-normal distribution is a good fit). The decoder is defined using the multinomial distribution model (words in the document are drawn from the same categorical distribution corresponding to its topic):

$$p(x, l \,|\, z) = p(l)h(x \,|\, l) \exp(f(z)^{\mathsf{T}} x - lA(f(z))), \tag{13}$$

where $f(z)$ is a neural network mapping the latent code to the logits of word occurrence probabilities, $A(\eta) = \log \sum_k \exp(\eta_k)$ and $h(x \,|\, l) = [\![\sum_k x_k{=}l]\!] \big(l! / \prod_k x_k!\big)$ is the base measure[2]. The sufficient statistics are the word counts $x$. The prior $p(z)$ is assumed uniform Bernoulli.

The encoder is the conditionally independent Bernoulli model, expressed as

$$q(z \,|\, x, l) \propto \exp(g(x)^{\mathsf{T}} z), \tag{14}$$

where $g(x)$ is the encoder network. Chaidaroon & Fang (2017) experimented with the encoder and decoder design and recommended using TFIDF features instead of raw counts. First, we note that the inverse document frequency (IDF) is not relevant, since it can be learned by the first linear transform in the encoder. Effectively, the term frequency (TF), given by $x/l$, is used. This choice is adopted in later works (Shen et al., 2018; Zamani Dadaneh et al., 2020; Ñanculef et al., 2020). It might seem reasonable that the latent code modeling the document topic should not depend on the document length, only on the distribution of words in the document. However, we will argue that this rationale is misleading for stochastic encoders.

We apply Theorem 1 to two groups of variables: observed $(x, l)$ and latent $z$ with $h(x, l) = h(x \,|\, l)p(l)$, $\nu(x, l) = x$ and $\psi(z) = z$. It follows that the consistent joint family is

$$p(x, l, z) = h(x, l) \exp(x^{\mathsf{T}} Wz + a^{\mathsf{T}} x + b^{\mathsf{T}} z + c). \tag{15}$$

This however implies that $g(x) = Wx + b$, *i.e.* the encoder network must be linear in $x$. Consequently, it cannot match a function of word frequencies $x/l$ (as chosen by design) unless $W = 0$, *i.e.* a completely trivial model with an encoder not depending on $x$. Such an encoder would imply full posterior collapse. The corresponding consistent set $\Theta_\Phi$ coincides with the set of collapsed VAEs where the decoder does not depend on the latent variable $z$ in Fig. 1. We conjecture that the inherent inconsistency of this VAE has a detrimental effect on learning.

If instead, we let the encoder network to access word counts $x$ directly, we obtain that $g(x) = Wx + b$ can form a consistent VAE. Inspecting this encoder model in more detail, we see that it builds up topic confidence in proportion to the evidence (total word counts), as the true posterior would. Indeed, the true posterior $p(z \,|\, x, l)$ satisfies the factorization by Bayes's theorem: $p(z \,|\, x, l) = p(x \,|\, z, l)p(z)/p(x \,|\, l)$. The prior $p(z)$ is constant by design, $p(x \,|\, l)$ does not vary with $z$ and $p(x \,|\, z, l)$ factors over all word instances according to (13). In other words, the coupling between $x$ and $z$ in $\log p(z \,|\, x, l)$ is linear in $x$.

In Section 4 we study the proposed correction experimentally and show that it enables learning better models under a variety of settings.

## 4 EXPERIMENTS

### 4.1 ARTIFICIAL EXAMPLE

To start with, we illustrate our findings on a toy example. We consider a simple Gaussian mixture model for which we can easily generate samples and compute all necessary quantities including the ELBO objective. We define the ground truth model to be $p^*(x, z) = p^*(z)p^*(x \,|\, z)$, with $z \in \{1 \ldots 4\}$, $p^*(z) \equiv 0.25$, $x \in \mathbb{R}^2$, $p^*(x \,|\, z) = \mathcal{N}(x \,|\, \mu(z), \sigma^2 I)$, *i.e.* a mixture of four 2D Gaussians. Fig. 2(a) shows the color-coded posterior distribution $p^*(z \,|\, x)$. We assign a color to each component and represent $p^*(z \,|\, x)$ for each pixel $x \in \mathbb{R}^2$ by the corresponding mixture of the component colors. For better interpretability, we illustrate further results by decision maps $\arg\max_z p(z \,|\, x)$. Fig. 2(b) shows the decision map for $p^*(z \,|\, x)$.

---

[2]Prior work (Chaidaroon & Fang, 2017; Shen et al., 2018) omits the base measure as it has no trainable parameters.

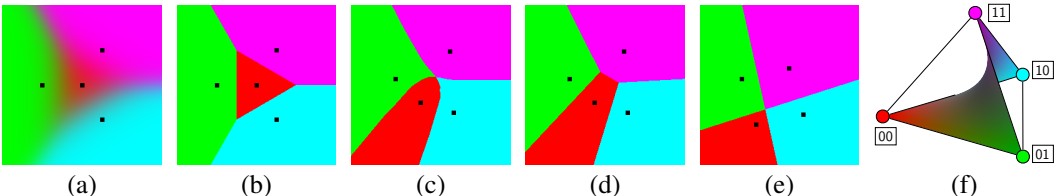

Figure 2: Artificial example. **(a)** Color-coded posterior distribution $p^*(z|x)$. (b-e): Decision maps $(\arg\max_z)$ of: **(b)** true posterior $p^*(z|x)$, **(c)** factorized encoder $q_\phi(z|x)$ after joint learning, **(d)** model posterior $p_\theta(z|x)$ after joint learning, **(e)** RBM trained on the same data. Gaussian centers $\mu(z)$ are shown as black dots. **(f)** Probability simplex of distributions over the four binary configurations. The vertices correspond to pure (deterministic) binary states represented by the code and its respective color. The surface shows the manifold of factorized distributions realizable by $q(z|x)$. Notice that the two edges $(00, 11)$ and $(01, 10)$ are not in the manifold because they correspond to switching of two bits simultaneously in a correlated way. The factorized approximation cannot model transitions between these states. Hence, when learning VAE, these pairs of states are repulsed in the decision maps (c), (d).

The aim of the experiment is to learn a VAE and to study the influence of the factorization assumption on the results. We use the decoder architecture as in the ground truth model — a Gaussian distribution $p_\theta(x|z) = \mathcal{N}(x|\theta z_{oh}, \sigma^2 I)$, where $z_{oh}$ is the one-hot (categorical) representation of $z$, and $\theta$ is a $2{\times}4$ matrix that maps the four latent codes to 2D centers at general locations. Note that the ground truth model is contained in the chosen decoder family. Hence, the ML solution is the ground truth decoder $p^*(x|z)$. We restrict the encoder to factor over the binary representation of the code $z_b \in \{0,1\}^2$ and define it as $q_\phi(z_b|x) \propto \exp\langle g_\phi(x), z_b\rangle$, where $g_\phi(x)$ is implemented as a feed-forward network with two hidden layers, each with 64 units and ReLU activations.

First, we pre-train our factorized encoder by optimizing ELBO and keeping the ground truth decoder fixed. The next step is to jointly train the encoder and decoder by maximizing ELBO. Since we are interested how the ELBO objective distorts the likelihood solution, we start with the ground truth decoder and the pre-trained encoder from the previous step. The ELBO-optimal decoder has to match not only the training data, but also the inexact, factorizing encoder. The resulting $q_\phi(z|x)$ is shown in Fig. 2(c) and the learned model posterior $p_\theta(z|x) \propto p(z)p_\theta(x|z)$ in Fig. 2(d). Note that they match each other pretty well, but differ substantially from the ground truth posterior shown in Fig. 2(b). The impact of the factorization is clearly visible – one can see two decision boundaries (one for each bit of $z_b$), which together partition the $x$-space into four regions, approximating the true posterior. For comparison, Fig. 2(e) shows the posterior of an RBM trained on the same data. It is clearly seen that the ELBO optimizer is pulled away from the likelihood optimizer towards an RBM solution. Notice also the explanation given in Fig. 2(f). Summarizing, this simple toy example clearly shows the VAE approximation error caused by the combination of ELBO objective and the factorization assumption for the encoder. While the numerical difference between ELBO and log-likelihood is small (see details in Appendix C.1), the qualitative difference in Fig. 2 appears substantial.

## 4.2 GAUSSIAN VAES FOR CELEBA IMAGES

The goal and design of this experiment is similar to the previous one. We first define a ground truth decoder which is used to generate training images. Then we pre-train an encoder by ELBO keeping the ground truth decoder fixed. Finally, we train both model parts starting from the ground truth decoder and the pre-trained encoder.

The ground truth generative model is obtained by training a convolutional Generative Adversarial network (GAN) using code of Inkawhich (2017) on the CelebA dataset Liu et al. (2015). We scale and crop all images to $64{\times}64$ pixels. In order to get a stochastic decoder, we equip the GAN generator $x = d(z)$, $z \in \mathbb{R}^{100}$, $x \in \mathbb{R}^{64 \times 64 \times 3}$ with *image noise* $\sigma_d$. The ground truth generative model is thus defined as $p^*(x, z) = p^*(z)p^*(x|z)$, where $p^*(z) = \mathcal{N}(z|0, I)$, $p^*(x|z) = \mathcal{N}(x|\mu_d(z), \sigma_d^2 I)$, and $\sigma_d^2$ is a common noise variance for all pixels and color channels. We chose $\sigma_d = 0.05$ (the color values are normalized to $[-1, 1]$). This corresponds to an image noise level, which is just visible, but does not disturb visual perception essentially.

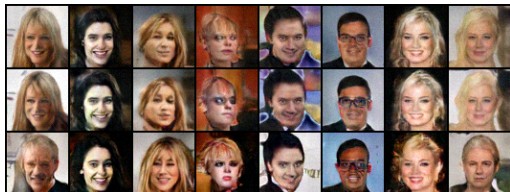 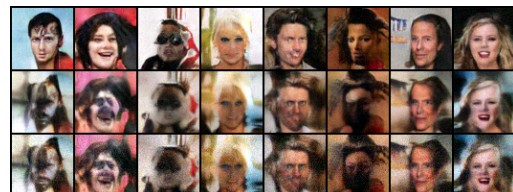

Figure 3: Results for the encoder learned by supervised conditional likelihood. Top row: training samples $\hat{x} \sim p^*(x)$. Second row: the corresponding reconstructions from mean values of $z$, i.e. $x \sim p^*(x \,|\, \mu_e(\hat{x}))$. Third row: reconstructions from sampled $z$, i.e. $\hat{z} \sim \mathcal{N}(\mu_e(\hat{x}), \sigma_e^2(\hat{x}))$ followed by $x \sim p^*(x \,|\, \hat{z})$.

Figure 4: Visual comparison – images drawn from the original/learned models. Each column corresponds to a particular value of $z \sim \mathcal{N}(0, I)$. Top row: the ground truth model, middle row: learned decoder with fixed $\sigma_d$, bottom row: decoder with learned $\sigma_d$.

The decoder family of the considered VAE consists of networks with the same architecture as $d(z)$. This ensures that the ground truth decoder is a likelihood maximizer of the VAE model. The encoder is defined as $q_\phi(z \,|\, x) = \mathcal{N}(x \,|\, \mu_e(x), \mathrm{diag}(\sigma_e^2(x)))$, where $\mu_e, \sigma_e \in \mathbb{R}^{100}$ are two outputs of a convolutional neural network with an architecture similar to the architecture of the discriminator used for training the GAN (except the output layer), *i.e.*, $q_\phi(z \,|\, x)$ is a multivariate Gaussian with diagonal covariance matrix whose parameters depend on $x$.

We pre-train the encoder fully supervised by maximizing its conditional log-likelihood $\mathbb{E}_{p^*(x,z)} \log q_\phi(z \,|\, x)$ on examples drawn from the ground truth generating model $p^*(x, z)$.[3] The results of pre-training are shown in Fig. 3. Then we jointly learn the encoder and decoder by maximizing ELBO on $x$-samples drawn from the ground truth model. We start the learning with the ground truth decoder $p^*(x \,|\, z)$ and the encoder obtained in the previous step. Two variants are considered for this training: (i) keeping the image noise $\sigma_d$ fixed and (ii) learning it along with other model parameters. We evaluate the results quantitatively by computing the Frechet Inception Distances (FID) between the ground truth model $p^*(x \,|\, z)$ and the obtained decoders $p_\theta(x \,|\, z)$ using the code of Seitzer (2020). For this we generate 200k images from each model. The obtained values are given in Tab. 1. Fig. 4 shows images generated by the ground truth model and the two learned models.

To conclude, ELBO optimization harms the decoder considerably as clearly seen both from FID-scores and the generated images. Models with higher ELBO values have worse FID-scores and produce less realistic images.

Table 1: Optimizing the ELBO starting from the ML solution degrades the FID-score. The first row corresponds to the pre-trained encoder for the ground truth decoder, its FID-score therefore compares two image sets, both generated by the ground truth model.

| Experiment | ELBO | FID |
|---|---|---|
| optimize encoder (conditional likelihood) | -364513.78 | 0.13 |
| optimize encoder and decoder (ELBO, fixed $\sigma_d$) | -5898.94 | 77.10 |
| optimize encoder and decoder (ELBO, learned $\sigma_d$) | 9035.69 | 117.87 |

### 4.3 BERNOULLI VAE FOR TEXT DOCUMENTS

This experiment compares training of the VAE model for semantic hashing discussed in Section 3.1 with and without our proposed correction on the *20Newsgroups* dataset (Lang & Rennie, 2008). We describe the dataset, preprocessing and optimization details in Appendix C.2.

We compare three encoders: **e1**: linear encoder on word counts (the proposed correction), **e2**: deep (2 hidden layers) encoder using word frequencies and **e3**: a linear encoder on frequencies. The

---

[3] In contrast to the previous experiment we optimize the "forward" KL-divergence, i.e. the conditional likelihood, instead of the "reverse" KL-divergence used in ELBO for simplicity.

encoders are compared across different numbers of latent Bernoulli variables (bits) and different decoder depths. The decoder depth denotes the number of fully connected hidden ReLU layers (0-2). In both the encoder and decoder we use 512 units in hidden layers. The prior work mainly used linear decoders following the ablation study of Shen et al. (2018). Our experiments also suggest that using deep decoders in combination with longer bit-length leads to a significant overfitting. When the decoder is linear, the posterior distribution is tractable and is linear as well, *i.e.*, the VAE model is equivalent to a Multinomial-Bernoulli RBM. We experimentally verify that a linear encoder on word counts e1 indeed works better in this case. However, perhaps more surprisingly, we also find out that it works better even for non-linear decoders.

Table 2 show the achieved training and test Negative ELBO (NELBO) values. We observe across all settings that the simple encoder e1 is consistently better than the more complex encoder e2 which in turn is significantly better than the linear encoder on frequencies e3. We conclude that the use of VAEs with deep encoders based on word frequencies (Chaidaroon & Fang, 2017; Shen et al., 2018; Zamani Dadaneh et al., 2020; Ñanculef et al., 2020) is sub-optimal for this dataset. We also observe that linear *decoders* generalize better under 32 and 64 bits compared to more complex decoders, which suffer from overfitting. This implies that in these cases the best encoder-decoder combination is linear, *i.e.* the basic RBM model. This evidence agrees with previously observed worse reconstruction error with deep architecture (Dai et al., 2020), however we did not observe (a more severe) posterior collapse with deeper models amongst the depths we report.

Table 2: Training and test NELBO values for Text-VAE with different configurations of bits, decoder and encoder. Bold highlights the best encoder choice and underlined bold values are the best decoder-encoder combinations for each bit-length.

| | TRAINING | | | | | | | | | | TEST | | | | | | | | |
|---|---|---|---|---|---|---|---|---|---|---|---|---|---|---|---|---|---|---|---|
| **Bits** | dhidden=0 | | | dhidden=1 | | | dhidden=2 | | | **Bits** | dhidden=0 | | | dhidden=1 | | | dhidden=2 | | |
| | e1 | e2 | e3 | e1 | e2 | e3 | e1 | e2 | e3 | | e1 | e2 | e3 | e1 | e2 | e3 | e1 | e2 | e3 |
| 8 | **419** | 429 | 439 | **321** | 390 | 415 | **325** | 370 | 421 | 8 | **423** | 429 | 435 | **413** | 421 | 424 | **418** | 423 | 427 |
| 16 | **382** | 398 | 419 | **201** | 329 | 407 | **164** | 269 | 413 | 16 | **409** | 417 | 421 | **404** | 422 | 420 | **410** | 416 | 422 |
| 32 | **337** | 358 | 412 | **165** | 189 | 403 | **132** | 159 | 411 | 32 | **396** | 413 | 416 | **399** | 413 | 418 | **406** | 416 | 421 |
| 64 | **296** | 324 | 407 | **171** | 189 | 398 | **134** | 149 | 409 | 64 | **392** | 411 | 414 | **398** | 413 | 417 | **406** | 417 | 422 |

## 5 CONCLUSIONS

We have analyzed the approximation error of VAEs in a general setting, when both the decoder and encoder are exponential families. This includes commonly used VAE variants as, *e.g.*, Gaussian VAEs and Bernoulli VAEs. We have shown that the subset of generative models consistent with the encoder class is quite restricted: it coincides with the set of log-bilinear models on the sufficient statistics of both decoder and encoder, *i.e.*, RBM-like models. This consistent subset can not be enlarged by using more complex encoder networks as long as encoder's sufficient statistics remain unchanged. In combination with the ELBO objective, this causes an approximation error — the ELBO optimizer is pulled away from the data likelihood optimizer towards this subset. Moreover, we proved theoretically that close-to-tight EF VAEs must be close to RBMs in a certain sense.

We have shown that the error is detrimental when the consistent subset is too restrictive. In the cases where a lot of data is available and a high quality generative model is of the primary interest, such as in the CelebA experiment, more expressive encoder families are required in addition to large networks. On the other hand the VAE approximation error may result in a useful regularization when the respective RBM is a good baseline model. In this case we can speak of a binning inductive bias towards RBM, such as in our text-VAE experiment. Furthermore, simple encoders can be desired when the learned representations are of interest, in particular they appear to facilitate similarity in Hamming distance, useful in the semantic hashing problem.

Further connections to related work and discussion can be found in Appendix B.

ACKNOWLEDGMENT

D.S. was supported by the German Federal Ministry of Education and Research (BMBF, 01/S18026A-F) by funding the competence center for Big Data and AI "ScaDS.AI Dresden/Leipzig". A.S and B.F gratefully acknowledge support by the Czech OP VVV project "Research Center for Informatics" (CZ.02.1.01/0.0/0.0/16019/0000765)". B.F. was also supported by the Czech Science Foundation, grant 19-09967S. The authors gratefully acknowledge the Center for Information Services and HPC (ZIH) at TU Dresden for providing computing time. We thank the anonymous reviewers for many helpful links and suggestions.

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

# Appendix

## A PROOFS

### A.1 PROOF OF THEOREM 1

The proof directly follows from the characterization of conditionally specified joint distributions in the exponential family given by Arnold & Strauss (1991), see (Arnold et al., 2001, Theorem 3):

**Theorem A.1** (Arnold & Strauss 1991). *Let $x \in \mathcal{X}$ and $z \in \mathcal{Z}$ be random variables with a strictly positive joint distribution such that both conditional distributions are exponential families with densities*

$$p(x \mid z) = h(x) \exp\left[\langle \nu(x), f(z) \rangle - A(z)\right] \tag{16a}$$

$$p(z \mid x) = h'(z) \exp\left[\langle \psi(z), g(x) \rangle - B(x)\right], \tag{16b}$$

*where $\nu : \mathcal{X} \to \mathbb{R}^n$ and $\psi : \mathcal{Z} \to \mathbb{R}^m$ are minimal sufficient statistics, $f : \mathcal{Z} \to \mathbb{R}^n$ and $g : \mathcal{X} \to \mathbb{R}^m$ are any mappings, $h(x)$ and $h'(z)$ are base measures, and $A$ and $B$ denote the respective log-partition functions[4].*

*Then there exists a matrix $W \in \mathbb{M}(n+1, m+1)$, such that the density of the joint distribution can be represented as*

$$p(x, z) = h(x)h'(z) \exp \langle \nu_e(x), W \psi_e(z) \rangle, \tag{17}$$

*where $\nu_e$, $\psi_e$ denote the statistics vectors extended with an additional component 1.*

### A.2 EXAMPLE OF A DISCRETE VAE APPROACHING A FLOW

**Example A.1.** In this example we construct a VAE that can be arbitrary close to tight one, but where the decoder network does not approach a linear map. Let $\mathcal{X} = \{-1, 1\}^2$, $\mathcal{Z} = \{-1, 1\}^2$ and let $p(z)$ be uniform. Let the decoder be conditionally independent

$$p(x \mid z) = \exp\left[\beta \langle x, \pi(z) \rangle - A(z)\right] \tag{18}$$

where $\pi(z)$ denotes the invertible mapping of $\mathcal{Z}$ to $\mathcal{X}$ given by

$$x_1 = z_1 \tag{19}$$

$$x_2 = z_1 z_2. \tag{20}$$

If the parameter $\beta$ is sufficiently large, the distribution $p(x \mid z)$ approaches the deterministic distribution $\delta_{x=\pi(z)}$. Its posterior therefore also approaches the deterministic distribution $\delta_{z=\pi^{-1}(x)}$ (notice that $\pi^{-1} = \pi$). Let the encoder be the conditionally independent model $q(z \mid x) = \exp\left[\beta \langle z, \pi^{-1}(x) \rangle - A(x)\right]$. This decoder by design approaches $\delta_{x=\pi(z)}$ as well and thus the VAE $(p, q)$ achieves $\varepsilon$-tightness for sufficiently large $\beta$. At the same time the deviation between logarithms of probabilities $\log q(z \mid x)$ and $\log p(z \mid x)$ grows with $\beta$.

### A.3 PROOF OF THEOREM 2

The idea of the proof is to bound the difference between $\log p(z \mid x)$ and $\log q(z \mid x)$, which is done by Proposition A.2 and then in the space of log-probabilities to approximate the non-linear mapping $f_e(z)$ by a linear one as detailed in Proposition A.1. By carefully choosing the norms and the approximation we obtain a bound on the error for the joint model, which despite the discreteness assumption of the observation space $\mathcal{X}$ in Theorem 2 does not depend on its cardinality.

For a finite set $X \subset \mathcal{X}$ let $\mathcal{H}$ be the $|X|$-dimensional vector space with the inner product $\langle u, v \rangle_{p_d} = \sum_{x \in X} p_d(x) u(x) v(x)$, assuming that $p_d(x) > 0$ for all $x \in \mathcal{X}$. The respective norm will be denoted as $\| \cdot \|_{p_d}$.

---

[4]The log-partition function is usually defined as a function of the (natural) parameter. In this context we consider $h, h', f, g$ to be fixed and consider the dependence on $x, z$ only.

**Proposition A.1.** *Under model Assumption 1, for any finite $X \subseteq \mathcal{X}$ there exists a matrix $W \in \mathbb{M}(n+1, m+1)$ such that joint distribution implied by the decoder $p(x, z) = p(x|z)p(z)$ can be approximated by an unnormalized EF Harmonium*

$$\tilde{p}(x, z) = h(x)h'(z)\exp(\langle \nu_e(x), W\psi_e(z)\rangle) \tag{21}$$

*with the error bound*

$$(\forall z) \sum_{x \in X} p_d(x)\big|\log p(x, z) - \log \tilde{p}(x, z)\big|^2 \leq \sum_{x \in X} p_d(x)\big|\log q(z|x) - \log p(z|x)\big|^2. \tag{22}$$

The function $\tilde{p}(x, z)$ is non-negative but does not necessarily satisfy the normalization constraint of a density.

*Proof.* For clarity, we will omit the dependence of the decoder and encoder on their parameters $\theta$, resp. $\phi$. Throughout the proof we will also assume that a single $z \in \mathcal{Z}$ is fixed.
First, we expand

$$\log q(z|x) = \langle \psi(z), g(x)\rangle - B(x) + \log h'(z); \tag{23a}$$
$$\begin{aligned}\log p(z|x) &= \log p(x|z) + \log p(z) - \log p(x) \\ &= \langle \nu(x), f(z)\rangle - A(z) + \log h(x) + \log p(z) - \log p(x),\end{aligned} \tag{23b}$$

where $A(z) = A(f(z))$ and $B(x) = B(g(x))$. We can therefore represent

$$\log q(z|x) - \log p(z|x) = \langle \psi(z), g(x)\rangle - B(x) + \log p(x) - \log h(x) \tag{24}$$
$$- \Big(\langle \nu(x), f(z)\rangle + \log p(z) - \log h'(z) - A(z)\Big) \tag{25}$$
$$= \langle \psi_e(z), g_e(x)\rangle - \langle \nu_e(x), f_e(z)\rangle, \tag{26}$$

where

$$\psi_e(z) = (\psi(z), \ 1); \tag{27}$$
$$\nu_e(x) = (\nu(x), \ 1); \tag{28}$$
$$g_e(x) = (g(x), \ \log p(x) - B(x) - \log h(x)); \tag{29}$$
$$f_e(z) = (f(z), \ \log p(z) - A(z) - \log h'(z)). \tag{30}$$

With this representation we have:

$$\sum_{x \in X} p_d(x)\big|\langle \nu_e(x), f_e(z)\rangle - \langle \psi_e(z), g_e(x)\rangle\big|^2 \tag{31}$$
$$= \sum_{x \in X} p_d(x)\big|\log q(z|x) - \log p(z|x)\big|^2 =: \Delta^2. \tag{32}$$

Let $V$ be the matrix with rows $\nu_e(x)$ for all $x \in X$. Let $G$ be the matrix with rows $g_e(x)$ for all $x \in X$. We can rewrite the condition (31) in the form

$$\xi = V f_e(z) - G\psi_e(z), \tag{33a}$$
$$\|\xi\|_{p_d}^2 = \Delta^2, \tag{33b}$$

where $\xi \in \mathbb{R}^{|X|}$ is the vector of residuals. Let $P$ be the orthogonal projection onto the range of $V$ in the space $\mathcal{H}$. Multiplying (33a) by $P$ on the left, we obtain

$$PV f_e(z) - PG\psi_e(z) = P\xi. \tag{34}$$

Because $PV = V$ we obtain

$$V f_e(z) - PG\psi_e(z) = P\xi. \tag{35}$$

We therefore can consider the decoder network approximation $\bar{f}_e(z) = \tilde{W}\psi_e(z)$, where $\tilde{W}$ is the solution to the consistent system of linear equations $V\tilde{W} = PG$ and is independent of $z$. We can therefore express

$$\|V f_e(z) - V \bar{f}_e(z)\|_{p_d} = \|P\xi\|_{p_d} \leq \|\xi\|_{p_d} = \Delta, \tag{36}$$

where the inequality holds because $P$ is an orthogonal projection in $\mathcal{H}$.

We obtained that the decoder network $f_e(z)$ can be approximated by a linear mapping $\tilde{W}\psi_e(z)$ such that

$$\sum_{x \in X} p_d(x) \big| \langle \nu_e(x), f_e(z) \rangle - \langle \nu_e(x), \tilde{W}\psi_e(z) \rangle \big|^2 \le \Delta^2. \tag{37}$$

Expressing back

$$\langle \nu_e(x), f_e(z) \rangle = \langle \nu(x), f(z) \rangle + \log p(z) - A(z) - \log h'(z) \tag{38a}$$
$$= \log p(x \,|\, z) + \log p(z) - \log h(x) - \log h'(z) \tag{38b}$$
$$= \log p(x, z) - \log h(x) - \log h'(z) \tag{38c}$$

we obtain that

$$\sum_{x \in X} p_d(x) \big| \log p(x, z) - \log \tilde{p}(x, z) \big|^2 \le \Delta^2, \tag{39}$$

where

$$\log \tilde{p}(x, z) = \log h(x) + \log h'(z) + \langle \nu_e(x), W\psi_e(z) \rangle. \tag{40}$$

$\square$

**Proposition A.2.** *Under model Assumption 1, let $\mathcal{X}$ and $\mathcal{Z}$ be discrete (finite) sets and let $z \in \mathcal{Z}$ be chosen. If $q(z \,|\, x) \ge \alpha$ and $p(z \,|\, x) \ge \alpha$ for all $x \in X$, where $X \subseteq \mathcal{X}$ and*

$$\mathbb{E}_{p_d(x)}[D_{\mathrm{KL}}(q(z \,|\, x) \,\|\, p(z \,|\, x))] \le \varepsilon, \tag{41}$$

*then*

$$\sum_{x \in X} p_d(x) \big( \log p(z \,|\, x) - \log q(z \,|\, x) \big)^2 \le \frac{\varepsilon}{\alpha^2} + o(\varepsilon). \tag{42}$$

*Proof.* Let us denote $\varepsilon(x) = D_{\mathrm{KL}}(q(z \,|\, x) \,\|\, p(z \,|\, x))$. Pinsker's inequality assures for each $x$

$$\sup_{S \subset \mathcal{Z}} |\mathbb{P}_{q(z \,|\, x)}(S) - \mathbb{P}_{p(z \,|\, x)}(S)|^2 \le \varepsilon(x)/2. \tag{43}$$

Substituting $S = \{z\}$ we obtain

$$|p(z \,|\, x) - q(z \,|\, x)|^2 \le \varepsilon(x)/2. \tag{44}$$

By taking expectation in $p_d(x)$ on both sides we obtain a variant of Pinsker's inequality:

$$\mathbb{E}_{p_d(x)} |p(z \,|\, x) - q(z \,|\, x)|^2 \le \mathbb{E}_{p_d(x)} \varepsilon(x)/2 = \tfrac{1}{2} \mathbb{E}_{p_d(x)}[D_{\mathrm{KL}}(q(z \,|\, x) \,\|\, p(z \,|\, x))] \le \varepsilon/2. \tag{45}$$

Notice that the LHS depends on the given $z$. Because all summands are non-negative it follows that

$$\forall X' \subseteq \mathcal{X} \quad \sum_{x \in X'} p_d(x) |p(z \,|\, x) - q(z \,|\, x)|^2 \le \varepsilon/2. \tag{46}$$

This inequality will be used in several places below.

Consider $x \in X$ such that $p(z \,|\, x) > q(z \,|\, x)$. Then

$$|\log p(z \,|\, x) - \log q(z \,|\, x)|^2 = \log^2 \tfrac{p(z \,|\, x)}{q(z \,|\, x)} \tag{47a}$$
$$= \log^2 (1 + \tfrac{p(z \,|\, x) - q(z \,|\, x)}{q(z \,|\, x)}) \tag{47b}$$
$$\le \log^2 (1 + \tfrac{p(z \,|\, x) - q(z \,|\, x)}{\alpha}), \tag{47c}$$

where the inequality holds because $\log^2$ is monotonously increasing for arguments greater equal than 1, which is ensured. Let us now consider $x \in X$ such that $p(z \,|\, x) < q(z \,|\, x)$. Then

$$|\log p(z \,|\, x) - \log q(z \,|\, x)|^2 = \log^2 \tfrac{q(z \,|\, x)}{p(z \,|\, x)} \tag{48a}$$
$$= \log^2 (1 + \tfrac{q(z \,|\, x) - p(z \,|\, x)}{p(z \,|\, x)}) \tag{48b}$$
$$\le \log^2 (1 + \tfrac{q(z \,|\, x) - p(z \,|\, x)}{\alpha}). \tag{48c}$$

In total, we obtain

$$|\log p(z \,|\, x) - \log q(z \,|\, x)|^2 \le \log^2 (1 + \tfrac{|p(z \,|\, x) - q(z \,|\, x)|}{\alpha}). \tag{49}$$

Let us denote $u(x) = \frac{|p(z\,|\,x) - q(z\,|\,x)|}{\alpha}$ and partition the set $X$ into two parts:

$$X_1 = \{x \in X \mid u(x) < u_0\} \tag{50a}$$
$$X_2 = \{x \in X \mid u(x) \geq u_0\}, \tag{50b}$$

where we chose $u_0 \in [\sqrt{e-1}, 5]$ for reasons to be clarified below. For both parts, *i.e.* $k = 1, 2$, we have

$$\sum_{x \in X_k} p_d(x) |\log q(z\,|\,x) - \log p(z\,|\,x)|^2 \leq \sum_{x \in X_k} p_d(x) \log^2(1 + \frac{|p(z\,|\,x) - q(z\,|\,x)|}{\alpha}). \tag{51}$$

For $X_1$, (51) can be further bounded as

$$\leq \sum_{x \in X_1} p_d(x) \log(1 + \frac{|p(z\,|\,x) - q(z\,|\,x)|^2}{\alpha^2}) \tag{52a}$$
$$\leq \log \sum_{x \in X_1} p_d(x) \left(1 + \frac{|p(z\,|\,x) - q(z\,|\,x)|^2}{\alpha^2}\right) \tag{52b}$$
$$= \log \left(p_d(X_1) + \frac{1}{\alpha^2} \sum_{x \in X_1} p_d(x) |p(z\,|\,x) - q(z\,|\,x)|^2\right) \tag{52c}$$
$$\leq \log \left(1 + \frac{\varepsilon}{2\alpha^2}\right) = \frac{\varepsilon}{2\alpha^2} + o(\varepsilon), \tag{52d}$$

where the first inequality holds for $u_0 \leq 5$, because in this case $\log^2(1+u) \leq \log(1+u^2)$ holds, the second inequality is the Jensen's inequality for $\log$ and the last inequality uses (46) under monotone $\log$.

For $X_2$ we have the following. Let $V = p_d(X_2) = \sum_{x \in X_2} p_d(x)$. We can express

$$\sum_{x \in X_2} p_d(x) \log^2 \left(1 + \frac{|p(z\,|\,x) - q(z\,|\,x)|}{\alpha}\right) \tag{53a}$$
$$= V \left(\sum_{x \in X_2} p_d(x\,|\,X_2) \log^2 \left(1 + \frac{|p(z\,|\,x) - q(z\,|\,x)|}{\alpha}\right)\right) \tag{53b}$$

Using that $u < u^2$ on $X_2$ and that $\log^2(1 + u)$ is monotone for a positive argument, we can bound (53b) as

$$\leq V \left(\sum_{x \in X_2} p_d(x\,|\,X_2) \log^2 \left(1 + \frac{|p(z\,|\,x) - q(z\,|\,x)|^2}{\alpha^2}\right)\right). \tag{54}$$

Further, using that $\log^2(1 + v)$ is concave on $X_2$ for $v \geq e - 1$, we have

$$\leq V \log^2 \left(\sum_{x \in X_2} p_d(x\,|\,X_2)\left(1 + \frac{|p(z\,|\,x) - q(z\,|\,x)|^2}{\alpha^2}\right)\right) \tag{55a}$$
$$= V \log^2 \left(1 + \sum_{x \in X_2} p_d(x\,|\,X_2) \frac{|p(z\,|\,x) - q(z\,|\,x)|^2}{\alpha^2}\right) \tag{55b}$$
$$\leq V \log^2 \left(1 + \frac{\varepsilon}{2\alpha^2 V}\right), \tag{55c}$$

where in the last step we used (46).

Next we show that $V$ itself is bounded above by $\frac{\varepsilon}{2u_0^2 \alpha^2}$. It follows from

$$u_0^2 V = \sum_{x \in X_2} p_d(x) u_0^2 \leq \sum_{x \in X_2} p_d(x) \frac{|p(z\,|\,x) - q(z\,|\,x)|^2}{\alpha^2} \leq \sum_{x \in X} p_d(x) \frac{|p(z\,|\,x) - q(z\,|\,x)|^2}{\alpha^2} \leq \frac{\varepsilon}{2\alpha^2}, \tag{56}$$

where the last inequality is again (46).

Now, letting $r = \frac{\varepsilon}{2\alpha^2 V} \geq u_0^2 \geq 1$ we can bound (55c) as follows

$$V \log^2 \left(1 + \frac{\varepsilon}{2\alpha^2 V}\right) = \frac{\varepsilon}{2\alpha^2} \frac{1}{r} \log^2 \left(1 + r\right) \tag{57a}$$
$$\leq \frac{\varepsilon}{2\alpha^2} \sup_{r \geq 1} \frac{1}{r} \log^2 \left(1 + r\right) \leq \frac{\varepsilon}{2\alpha^2} 0.64. \tag{57b}$$

$\square$

Theorem 2 follows by Proposition A.2 and Proposition A.1 with the choice $X = \mathcal{X}$.

## B  DISCUSSION

In this section we discuss further connections to related work and some open questions.

Cremer et al. (2018) showed experimentally that using a more expressive class of models for the encoder reduces not only the posterior family mismatch but also the amortization error. This observation is compatible with our results: increasing the expressive power of the encoder admits tight VAEs with more complex dependence of $z$ on $x$. Indeed, increasing the expressive power of the encoder in our setting means extending its sufficient statistics $\psi(z)$ by new components. While it keeps the simple linear dependence $g(x) = W^\mathsf{T}\nu(x)$ characterizing tight VAEs, it does lead to a more expressive GLM $p(z\,|\,x)$. Conversely, our results suggest that increasing the complexity of the encoder network in order to reduce the amortization gap is only useful for models that are far from the consistent set. Furthermore, there could be negative impact from increasing the model depth in practice: (Dai et al., 2020) demonstrated that the risk of the learning converging to a suboptimal solution (in particular leading to more collapsed latent dimensions) increases with decoder depth.

Lucas et al. (2019) showed that any spurious local minima in linear Gaussian VAEs are entirely due to the marginal log likelihood and that the ELBO does not introduce any new local minima. It is a good question[5] whether something similar can be said about the EF VAE generalization. To our best knowledge this is not straightforward in such a general setting. The result of Lucas et al. (2019) is possible thanks to the fact that for linear Gaussian models ELBO is analytically tractable and its stationary point conditions can be written down and analyzed. In the general EF setup, which includes, *e.g.*, MRF VAEs, this does not appear possible. On the other hand, for any consistent EF VAE, the decoder posterior must be in the EF of the encoder. Therefore the optimal encoder could be easier to find analytically or numerically using forward KL divergence and not the reverse KL divergence used in ELBO, thus circumventing the question about local optima of ELBO. Since ELBO at the optimal encoder is tight for a consistent VAE, this could be an alternative way to find the global maximum.

A recent work by Sicks et al. (2021) extends the result of Lucas Lucas et al. (2019) in that they develop an analytical local approximation to ELBO, which is exact in the Gaussian linear model case and is a lower bound on ELBO for Binomial observation model. These results allow to analyze ELBO (and in particular the posterior collapse problem) locally under the assumption that the decoder's mapping $f(z)$ is (locally) an affine mapping of $z$. Our Theorem 1 implies it must be so globally for tight VAEs in several special cases (*e.g.*, Bernoulli model), while in general it is an affine mapping of $\psi(z)$. These connections indicate that a better understanding of VAEs can be reached in the setting where either decoder or encoder or both are consistent with the joint model (7).

We restricted this study to exponential families. While in richer models discussed in the introduction, the approximation error still exists, it is made small by design, and it is less relevant and harder to analyze it theoretically. One possible open direction where such analysis would make sense is to consider fully factorized non-exponential cases, *e.g.*, Student-t VAEs (Takahashi et al., 2018) or models satisfying hierarchical or partial factorization (Maaløe et al., 2019).

## C  DETAILS OF EXPERIMENTAL SETUP

### C.1  ARTIFICIAL EXAMPLE

We give here the achieved likelihood and ELBO values for this experiment. The negative entropy $\sum_x p^*(x) \log p^*(x)$ of the ground truth model, *i.e.*, the best reachable data log-likelihood, is $-3.65$. The ELBO of the pre-trained VAE is $-3.74$. During the second step of training, *i.e.*, the joint learning of the encoder and decoder by ELBO maximization, the ELBO value increases from $-3.74$ to $-3.70$. The data log-likelihood $\sum_x p^*(x) \log p_\theta(x)$ drops at the same time from $-3.65$ to $-3.68$. The approximation error (2) in the decoder caused by using the factorized encoder is only $0.03$ nats, but the qualitative difference between the ground truth model and the ELBO optimizer model shown in Fig. 2 appears detrimental.

---

[5]pointed out by reviewers

## C.2 BERNOULLI VAE FOR TEXT DOCUMENTS

**Dataset**    In this experiment we used the version of the *20Newsgroups* data set (Lang & Rennie, 2008) denoted as "processed" by the authors. The dataset contains bag-of-words representations of documents and is split into a training set with 11269 documents and a test set with 7505 documents. We keep only the 10000 most frequent words in the training set, which is a common pre-processing (each of the omitted words occurs not more than in 10 documents).

**Optimization**    To train VAE we used the state-of-the-art unbiased gradient estimator ARM (Yin & Zhou, 2019) and Adam optimizer with learning rate 0.001. We did not use the test set for parameter selection. We train for 1000 epochs using 1-sample ARM and then for 500 more epochs using 10 samples for computing each gradient estimate with ARM. We report the lowest negative ELBO (NELBO) values for the training set and test set during all epochs.

**Models**    The decoder model (13) describes words as independent draws from a categorical distribution specified by the neural network $f(z)$. This network respectively has a structure

$$\text{Linear} \rightarrow \underbrace{(\text{ReLU} \rightarrow \text{Linear})}_{\times \text{dhidden}} \rightarrow \text{Logsoftmax}.$$

The input dimension equals to the number of latent bits, the output dimension equals the number of words in the dictionary, 10000. For decoders with dhidden $= 1, 2$ the hidden layers contained 512 units.

The encoder networks **e2**, **e3** take on the input word frequencies $x / \sum_k x_k$, the encoder network **e1** takes on input word counts $x$. For the deep encoder **e2** we used 2 hidden ReLU layers with 512 units each.

