# OpenReview forum: "VAE Approximation Error: ELBO and Exponential Families"
_ICLR.cc/2022/Conference — ICLR 2022 Spotlight_

### Official Review · Reviewer_vibV · 2021-11-01

**Correctness:** 4
**Technical Novelty And Significance:** 3
**Empirical Novelty And Significance:** 4
**Recommendation:** 8
**Confidence:** 3

**Main Review:**

The paper studies the VAE approximation error, when the encoder and decoder distributions are from conditional exponential families (EFs). Using the definition of a consistent set Theta_{Phi} as above eq 3 (i.e. there exists a phi s.t. q_{phi}(z|x) \equiv p_{theta}(z|x)), Theorem 1 characterizes the general form of the joint probability (an EF-Harmonium as per eq 7) which satisfies the consistent set condition for exponential family VAEs. It is interesting that this is an *undirected* model, while VAEs are usually thought of a directed z->x latent variable model. This result is interesting, but as it is a direct consequence of a theorem by Arnold and Strauss (1991) [as cited] it clearly has less novelty than if it was derived by the authors.

It is shown that a small VAE posterior mismatch does not have to imply closeness of the optimal decoder to some decoder in the consistent set, as per the counterexample below Defn 1. However, by prohibiting nearly deterministic encoders and decoders, Theorem 2 can be proved where a small VAE posterior mismatch does imply closeness of the optimal decoder to some decoder in the consistent set.

Sec 3.1 presents specific examples of such EF-VAEs, namely the diagonal Gaussian VAE, the Bernoulli-MRF VAE, and the Bernoulli VAE for semantic hashing. In this last case, the form derived from the EF-VAE differs somewhat from the "standard" version, by using word counts rather than frequencies.

Sec 4 give some experimental results. Sec 4.1 shows how a semantic hashing representation of the posterior (i.e. two +1/-1 bits as opposed to a 4-dim one hot code) does induce an approximation error in the model.

Sec 4.2 carries out an interesting experiment, where data is generated from a GAN trained on the CelebA dataset (plus Gaussian noise in image space). Given the (z,x) pairs a decoder can be readily learned for this model. What is interesting is that optimization of the decoder and encoder parameters to optimize the ELBO degrades the FID performance. This is a nice demonstration of how optimizing the ELBO for such a model can harm the quality of the generator.

Sec 4.3 show show the proposed correction (using word counts rather than frequencies) obtains better ELBO scores.

Overall, I find that this paper provides a useful extension to knowledge around VAEs. It provides a new connection between tight EF-VAEs and RBM-like models (EF-Harmoniums), and shows that the consistent subset cannot be enlarged by using a more complex encoder network if the encoder's sufficient statistics remain unchanged.  The experiments in sec 4.2 show interesting results about the "unlearning" of the true generator when the ELBO is optimized, and sec 4.3 shows that a more careful modeling choice (using word counts rather than frequencies) inspired by Theorem 1 leads to better ELBO performance.

* Other points

p 3 footnote -- punctuate this footnote better.

p 4 "It turns out that the KL divergence has a pinhole .." -- "pinhole" is a bit of an odd choice of word here -- maybe "leak"?

p 6 Fig 2 -- to my eyes it was very hard to see the difference between the pink and orange colors -- please choose different colors to make this clearer.

**Summary Of The Paper:**

The paper studies the VAE approximation error, when the encoder and decoder distributions are from conditional exponential families (EFs). Theorem 1 characterizes the form of the joint probability (an EF-Harmonium) which is consistent, i.e. where q_{phi}(z|x) \equiv p_{theta}(z|x). Sec 4.2 shows a v interesting example of Gaussian VAEs for CelebA images, where it is shown that a VAE "unlearns" the ground truth solution (being pulled away from the likelihood optimizer towards the consistent subset). Sec 4.3 shows another interesting example, where a careful formulation of the model identifies that it should use word counts rather than frequencies, and this is demonstrated empirically (Table 2).

**Summary Of The Review:**

Overall, I find that this paper provides a useful extension to knowledge around VAEs. It provides a new connection between tight EF-VAEs and RBM-like models (EF-Harmoniums), and shows that the consistent subset cannot be enlarged by using a more complex encoder network if the encoder's sufficient statistics remain unchanged.  The experiments in sec 4.2 show interesting results about the "unlearning" of the true generator when the ELBO is optimized, and sec 4.3 shows that a more careful modeling choice (using word counts rather than frequencies) inspired by Theorem 1 leads to better ELBO performance.

---

### Official Review · Reviewer_GwqU · 2021-11-03

**Correctness:** 3
**Technical Novelty And Significance:** 3
**Empirical Novelty And Significance:** 3
**Recommendation:** 8
**Confidence:** 3

**Main Review:**

Strengths.
The paper identifies and analyzes the approximation error in a general and realistic instance of the VAE with exponential family encoder and decoder and have backed it up with real examples from the literature including VAEs with RBM encoders and have used the theory to suggest improvements in the example of semantic hashing.

The theoretical justification of the suggestion that with exponential family encoder/decoder VAEs increasing depth does not help when the model is consistent or close to consistent is quite interesting. Some past work has shown empirical evidence that suggests that in many restricted cases adding depth does not help. I am not, however, aware of a clear theoretical justification of this empirical fact in the literature for the exponential family.

Furthermore, it has been shown in (Dai and Wipf, Diagnosing and Enhancing VAE models, 2019) that with Gaussian output models one can construct VAE decoders as bijections when the manifold dimension is equal to the data dimension with the consequence that the decoder is not linear. This paper shows that with some limits on the densities to exclude deterministic solutions, the VAE solution remains close to the consistent solution which matches the posterior perfectly. Taken together the results imply that VAE solutions are either consistent or close to consistent when used with exponential family distributions under some constraints.

Weaknesses/Improvements/Questions

Are the sub-optimal results shown in the experiments a result of the factorization assumption or are they a general problem with the ELBO objective? One might interpret the results in table 1 as suggesting that the problem might lie with the ELBO as well since increasing the ELBO lowers the FID. My question is whether the poor FID results from ELBO pathologies such as unbounded likelihood when modeling with continuous models or is it something inherent to the ELBO? Or is it a combination of the factorized encoder plus the ELBO? I don't think this point is as well addressed in the paper (even though the ELBO is identified as the culprit in the paragraph above table 1, for instance).

Lucas et al (2019) suggest for the case of linear Gaussian models that any spurious local minima are entirely due to the marginal log likelihood and that the ELBO does not introduce any new local minima. Can something similar be said for the generalization to exponential family distributions?

Similarly the toy example explains quite well the limitations of the factorization assumption but I don't think it says much about the limitations of the ELBO. Perhaps the authors can comment on this.

As a minor point, it would have been nice to see implications of theorems 1 and 2 explored in the experiments to see that exponential family VAEs remain close to linear parameterized exponential models.

Clarity.
The paper is generally clear and the claims are easy to follow.

Correctness.
The arguments in the paper appear correct to me, but I have not checked the proof in the appendix very carefully.

Relation to Previous Work.
The main portion of the paper can be seen as an extension of the work on linear VAEs (Lucas et al) to exponential families. The paper presents a theoretical analysis of the empirically observed approximation error previously observed in the literature. The idea of poor approximation is not novel (e.g., Cremer et al, 2018, and acknowledged in the paper) but the theoretical explanation is of some interest.

**Summary Of The Paper:**

Summary.
This paper presents an analysis of the approximation error of VAE models when the encoder and decoder are from exponential families. They show that when the model is consistent (i.e., the encoder is able to match the posterior), the encoder and decoder distributions are exponential family distributions that are linearly parameterized. The paper also shows that barring pathological cases, when the model is tight the decoder distribution is close to a linearly parameterized exponential family distribution. In particular, this shows that in this case additional depth is not useful.

Despite the restriction to exponential family distributions, the paper shows a number of real instances of models that lie in this class including Gaussian VAEs and VAEs with RBM encoders. For Bernoulli VAEs for semantic hashing, the paper proposes and verifies improvements to a model presented in the literature.


**Summary Of The Review:**

Overall, I think this is a well-written paper with an argument for the limitations of an important class of models. I did not find some of the experiments as convincing as the rest of the paper.

---

> ### Author Response · Authors · 2021-11-12
> **Answer**
>
> Thanks a lot for the detailed feedback.
>
> > Some past work has shown empirical evidence that suggests that in many restricted cases adding depth does not help.
>
> We would be grateful if you could recall such past work(s).
>
> > Are the sub-optimal results shown in the experiments a result of the factorization assumption or are they a general problem with the ELBO objective?
>
> We would like to clarify that the factorization assumption is one potential limitation of ELBO and not an independent problem on its own. The experiment is designed so that to maximally exclude other problems with ELBO as follows:
>  * The training data is generated on-line from a GT model based on GAN in order to avoid overfitting. The likelihood is therefore bounded from above by the likelihood of the GT model and is finite because the GT model has a fixed observation noise. The ELBO is bounded from above by the likelihood.
> * The VAE is initialized with the GT decoder and first optimized over the encoder only in order to have the tightest lower bound. This alleviates problems with local minima.
> * The encoder architecture is chosen rich enough so that the main limiting factor would be the factorization assumption.
>
> When optimising ELBO, we see that the model moves away from the initial point and ELBO increases. This clearly shows that the global optimiser of the data likelihood is not a maximum of ELBO. If there was no tightness gap due to factorization, we would expect the decoder to stay at the GT model, resulting in same or similar ELBO value and FID score.
>
> > Lucas et al (2019) suggest for the case of linear Gaussian models that any spurious local minima are entirely due to the marginal log likelihood and that the ELBO does not introduce any new local minima. Can something similar be said for the generalization to exponential family distributions?
>
> Thanks for the question, it is not straightforward. If true, such result would significantly generalize the proof of Lucas et al (2019) (for linear Gaussian models ELBO is analytically tractable and its stationary point conditions can be analyzed). On the other hand, for any consistent EF VAE, the decoder posterior must be in the EF of the encoder and the optimal encoder [edit] could be easier to find analytically or numerically with forward KL [/edit] regardless of the local optima of ELBO. The ELBO at the optimal encoder is tight, i.e. matches the log-likelihood.
>
> A recent related work
> Sicks et al. "A Generalised Linear Model Framework for β-Variational Autoencoders based on Exponential Dispersion Families" 2021
> extends the result of Lucas et al (2019) in that they develop an analytical local approximation to the ELBO, which is exact in the Gaussian linear model case and is a lower bound on ELBO for Binomial observation model.
>
> > Similarly the toy example explains quite well the limitations of the factorization assumption but I don't think it says much about the limitations of the ELBO. Perhaps the authors can comment on this.
>
> Indeed, in this work we focus on analyzing and demonstrating the limitations of ELBO due to the EF assumption, which is the encoder factorization in this case. We design the experiment so that to exclude other limitations: the decoder family includes the GT model, the encoder architecture is large enough, we avoid problems of local minima by initializing at GT decoder, we avoid overfitting by drawing all samples online from the generative model.

---

> > ### Comment · Reviewer_GwqU · 2021-11-17
> > **Further response**
> >
> > >> We would be grateful if you could recall such past work(s).
> >
> > I am thinking here of Dai et al. [1], where they show in the experiments that increasing depth can lead to worse reconstruction error.
> >
> > Thanks for the clarifications regarding the experiment from section 4.2. That helps answer my questions about that experiment. I think this is a fine paper and I will raise my score accordingly.
> >
> > [1] Dai et al, The usual suspects? Reassessing blame for posterior collapse.

---

### Official Review · Reviewer_3pHE · 2021-11-04

**Correctness:** 4
**Technical Novelty And Significance:** 3
**Empirical Novelty And Significance:** 3
**Recommendation:** 8
**Confidence:** 4

**Main Review:**

This is a very insightful paper with strong theoretical and empirical results and good writing. When first reading the paper my expectations were low because the problem of VAE posterior collapse and degeneracy is an old and tired one, however, the solution proposed by the paper is very interesting and to my knowledge completely novel.

Theory:

The theoretical contribution is very insightful. This paper observes that when the encoder and decoder distributions belong to the exponential family, only general linear models satisfy ‘consistency’ or even approximate consistency. Even though Thm 1 is not new, bringing up this result in the context of VAE is a valuable contribution. I think Thm 2 is a very interesting new result. This theorem somewhat conclusively puts a rest to the debate of whether flexible encoder distribution is necessary when learning with ELBO, by showing that unless the VAE is a flow or if the decoder is very simple, a flexible encoder distribution is needed.

The paper also shows in Corollary 1 that a flexible encoder network cannot make up for the lack of flexibility in encoder distribution. I think this is an interesting result that provides a lot of insight in the discussion of VAE mode collapse.

As the authors acknowledge, the analysis has a short-coming in that VAE models that approach a flow are exempt from the impossibility results, but I think that this exception does not detract from the interest of the general result.

Experiments:

The experiments are an insightful addition to the theoretical analysis. In particular, I really like the experiment where even with pretrained near-optimal decoders and encoders, further training on the ELBO degrades the decoder. I think this is a very clear experiment to show that failure of VAEs is often not caused by optimization issues, but rather the inherent limitations of ELBO learning with inflexible encoder distribution.

Other comments:

The literature review is somewhat lackluster, as there are probably tens of refereed papers discussing similar issues. For example, the following paper also provides perspectives on posterior collapse.

InfoVAE: Balancing Learning and Inference in Variational Autoencoders

The Usual Suspects? Reassessing Blame for VAE Posterior Collapse

That being said, to my knowledge the perspective in this paper is novel, but a good related work discussion on the debate around this topic can significantly strengthen the paper.

Small issues:

In Eq.(4a,4b) the partition function should also depend on theta and phi, also small typo in (4b)


**Summary Of The Paper:**

This paper proposes theoretical and empirical results on why VAEs tend to fail when the encoder distribution is not flexible enough.

**Summary Of The Review:**

Even though the problem is an old one that has been well studied, the theoretical results provide a very refreshing and novel perspective. The experimental results are also precisely the right experiments to complement the theory.

---

### Official Review · Reviewer_8hFF · 2021-11-04

**Correctness:** 4
**Technical Novelty And Significance:** 3
**Empirical Novelty And Significance:** 2
**Recommendation:** 6
**Confidence:** 3

**Main Review:**

**presentation & related work**

Authors do a nice job in this aspect.

**novelty & insights**

The interpretation and insight provided in theorem 1 is an interesting addition to the literature on analyzing VAEs, and is novel to the best of my knowledge. The empirical novelty, on the other hand, seems limited, given that the main experiments are demonstrating how an inferior variational family may lead the generative model astray even when the generative distribution are optimally initialized.

**experiments & empirical results**

The paper scores adequately in this regard.

**technical correctness**

I read through the proof of Theorem 1 and proof of the cited reference. These results are correct.

**Summary Of The Paper:**

Author studies VAEs under the assumption that the decoding distribution p(x|z) and approximate posterior distribution q(z|x) are in the exponential family. They show that the consistent set for the decoding distribution -- the set of p(x|z) which results in p(z|x) = q(z|x) for all z -- has a log-bilinear form in x and z. Consequently, optimizing an ELBO with such forms of exponential family p(x|z) and q(z|x) could result in a deviation from the maximum likelihood solution theta for p_theta(x).

**Summary Of The Review:**

The paper is overall of good quality. While the empirical novelty is somewhat limited, theorem 1 is insightful.

---

### Decision · Program_Chairs · 2022-01-20

**Decision:**

Accept (Spotlight)

**Comment:**

Wide agreement from the reviewers.  Interesting theorems.  Empirical work illustrates the theory.
Claim and insight: failure of VAEs is caused by the inherent limitations of ELBO learning with inflexible encoder distribution.
Good discussion pointed out related work and insights from the experiments.